# Robust Bayesian Max-Margin Clustering

**Changyou Chen**[†]       **Jun Zhu**[‡]       **Xinhua Zhang**[♯]

[†]Dept. of Electrical and Computer Engineering, Duke University, Durham, NC, USA

[‡]State Key Lab of Intelligent Technology & Systems; Tsinghua National TNList Lab;

[‡]Dept. of Computer Science & Tech., Tsinghua University, Beijing 100084, China

[♯]Australian National University (ANU) and National ICT Australia (NICTA), Canberra, Australia

cchangyou@gmail.com; dcszj@tsinghua.edu.cn; xinhua.zhang@anu.edu.au

## Abstract

We present max-margin Bayesian clustering (BMC), a general and robust framework that incorporates the max-margin criterion into Bayesian clustering models, as well as two concrete models of BMC to demonstrate its flexibility and effectiveness in dealing with different clustering tasks. The *Dirichlet process max-margin Gaussian mixture* is a nonparametric Bayesian clustering model that relaxes the underlying Gaussian assumption of Dirichlet process Gaussian mixtures by incorporating max-margin posterior constraints, and is able to infer the number of clusters from data. We further extend the ideas to present *max-margin clustering topic model*, which can learn the latent topic representation of each document while at the same time cluster documents in the max-margin fashion. Extensive experiments are performed on a number of real datasets, and the results indicate superior clustering performance of our methods compared to related baselines.

## 1   Introduction

Existing clustering methods fall roughly into two categories. Deterministic clustering directly optimises some loss functions, while Bayesian clustering models the data generating process and infers the clustering structure via Bayes rule. Typical deterministic methods include the well known kmeans [1], nCut [2], support vector clustering [3], Bregman divergence clustering [4, 5], and the methods built on the very effective max-margin principle [6–9]. Although these methods can flexibly incorporate constraints for better performance, it is challenging for them to finely capture hidden regularities in the data, e.g., automated inference of the number of clusters and the hierarchies underlying the clusters. In contrast, Bayesian clustering provides favourable convenience in modelling latent structures, and their posterior distributions can be inferred in a principled fashion. For example, by defining a Dirichlet process (DP) prior on the mixing probability of Gaussian mixtures, *Dirichlet process Gaussian mixture models* [10] (DPGMM) can infer the number of clusters in the dataset. Other priors on latent structures include the hierarchical cluster structure [11–13], co-clustering structure [14], *etc*. However, Bayesian clustering is typically difficult to accommodate external constraints such as max-margin. This is because under the standard Bayesian inference designing some informative priors (if any) that satisfy these constraints is highly challenging.

To address this issue, we propose Bayesian max-margin clustering (BMC), which allows max-margin constraints to be flexibly incorporated into a Bayesian clustering model. Distinct from the traditional max-margin clustering, BMC is fully Bayesian and enables probabilistic inference of the number of clusters or the latent feature representations of data. Technically, BMC leverages the regularized Bayesian inference (RegBayes) principle [15], which has shown promise on *supervised* learning tasks, such as classification [16, 17], link prediction [18], and matrix factorisation [19], where max-margin constraints are introduced to improve the discriminative power of a Bayesian

model. However, little exploration has been devoted to the *unsupervised* setting, due in part to the absence of true labels that makes it technically challenging to enforce max-margin constraints. BMC constitutes a first extension of RegBayes to the unsupervised clustering task. Note that distinct from the clustering models using maximum entropy principle [20, 21] or posterior regularisation [22], BMC is more general due to the intrinsic generality of RegBayes [15].

We demonstrate the flexibility and effectiveness of BMC by two concrete instantiations. The first is *Dirichlet process max-margin Gaussian mixture* (DPMMGM), a nonparametric Bayesian clustering model that relaxes the Gaussian assumption underlying DPGMM by incorporating max-margin constraints, and is able to infer the number of clusters in the raw input space. To further discover latent feature representations, we propose the *max-margin clustering topic model* (MMCTM). As a topic model, it performs max-margin clustering of documents, while at the same time learns the latent topic representation for each document. For both DPMMGM and MMCTM, we develop efficient MCMC algorithms by exploiting data augmentation techniques. This avoids imposing restrictive assumptions such as in variational Bayes, thereby facilitating the inference of the true posterior. Extensive experiments demonstrate superior clustering performance of BMC over various competitors.

## 2 Regularized Bayesian Inference

We first briefly overview the principle of regularised Bayesian inference (RegBayes) [15].

The motivation of RegBayes is to enrich the posterior of a probabilistic model by incorporating additional constraints, under an information-theoretical optimisation formulation. Formally, suppose a probabilistic model has latent variables $\Theta$, endowed with a prior $p(\Theta)$ (examples of $\Theta$ will be clear soon later). We also have observations $\mathbf{X} := \{\mathbf{x}_1, \cdots, \mathbf{x}_n\}$, with $\mathbf{x}_i \in \mathbb{R}^p$. Let $p(\mathbf{X}|\Theta)$ be the likelihood. Then, posterior inference via the Bayes' theorem is equivalent to solving the following optimisation problem [15]:

$$\inf_{q(\Theta) \in \mathcal{P}} \mathrm{KL}(q(\Theta) \,||\, p(\Theta)) - \mathbb{E}_{\Theta \sim q(\Theta)} \left[\log p(\mathbf{X}|\Theta)\right] \tag{1}$$

where $\mathcal{P}$ is the space of probability distribution[1], $q(\Theta)$ is the required posterior (here and afterwards we will drop the dependency on $\mathbf{X}$ for notation simplicity). In other words, the Bayesian posterior $p(\Theta|\mathbf{X})$ is identical to the optimal solution to (1). The power of RegBayes stems in part from the flexibility of engineering $\mathcal{P}$, which typically encodes constraints imposed on $q(\Theta)$, *e.g.*, via expectations of some feature functions of $\Theta$ (and possibly the data $\mathbf{X}$). Furthermore, the constraints can be parameterised by some auxiliary variable $\boldsymbol{\xi}$. For example, $\boldsymbol{\xi}$ may quantify the extent to which the constraints are violated, then it is penalised in the objective through a function $U$. To summarise, RegBayes can be generally formulated as

$$\inf_{\boldsymbol{\xi}, q(\Theta)} \mathrm{KL}(q(\Theta) \,||\, p(\Theta)) - \mathbb{E}_{\Theta \sim q(\Theta)} \left[\log p(\mathbf{X}|\Theta)\right] + U(\boldsymbol{\xi})$$
$$\text{s.t.} \quad q(\Theta) \in \mathcal{P}(\boldsymbol{\xi}). \tag{2}$$

To distinguish from the standard Bayesian posterior, the optimal $q(\Theta)$ is called *post-data posterior*. Under mild regularity conditions, RegBayes admits a generic representation theorem to characterise the solution $q(\Theta)$ [15]. It is also shown to be more general than the conventional Bayesian methods, including those methods that introduce constraints on a prior. Such generality is essential for us to develop a Bayesian framework of max-margin clustering. Note that like many sophisticated Bayesian models, posterior inference remains as a key challenge of developing novel RegBayes models. Therefore, one of our key technical contributions is on developing efficient and accurate algorithms for BMC, as detailed below.

## 3 Robust Bayesian Max-margin Clustering

For clustering, one key assumption of our model is that $\mathbf{X}$ forms a latent cluster structure. In particular, let each cluster be associated with a latent projector $\boldsymbol{\eta}_k \in \mathbb{R}^p$, which is included in $\Theta$ and has prior distribution subsumed in $p(\Theta)$. Given any distribution $q$ on $\Theta$, we then define the compatibility score of $\mathbf{x}_i$ with respect to cluster $k$ by using the marginal distribution on $\boldsymbol{\eta}_k$ (as $\boldsymbol{\eta}_k \in \Theta$):

$$F_k(\mathbf{x}_i) = \mathbb{E}_{q(\boldsymbol{\eta}_k)} \left[\boldsymbol{\eta}_k^T \mathbf{x}_i\right] = \mathbb{E}_{q(\Theta)} \left[\boldsymbol{\eta}_k^T \mathbf{x}_i\right] . \tag{3}$$

For each example $\mathbf{x}_i$, we introduce a random variable $y_i$ valued in $\mathbb{Z}^+$, which denotes its cluster assignment and is also included in $\Theta$. Inspired by conventional multiclass SVM [7, 23], we utilize $\mathcal{P}(\boldsymbol{\xi})$ in RegBayes (2) to encode the max-margin constraints based on $F_k(\mathbf{x}_i)$, with the slack variable $\boldsymbol{\xi}$ penalised via their sum in $U(\boldsymbol{\xi})$. This amounts to our Bayesian max-margin clustering (BMC):

$$\inf_{\xi_i \geq 0, q(\Theta)} \mathcal{L}(q(\Theta)) + 2c \sum_i \xi_i \tag{4}$$

$$\text{s.t.} \quad F_{y_i}(\mathbf{x}_i) - F_k(\mathbf{x}_i) \geq \ell \, \mathbb{I}(y_i \neq k) - \xi_i, \quad \forall i, k$$

where $\mathcal{L}(q(\Theta)) = \mathrm{KL}(q(\Theta)\|p(\Theta)) - \mathbb{E}_{\Theta \sim q(\Theta)}[\log p(\mathbf{X}|\Theta)]$ measures the KL divergence between $q$ and the original Bayesian posterior $p(\Theta|\mathbf{X})$ (up to a constant); $\mathbb{I}(\cdot) = 1$ if $\cdot$ holds true, and $0$ otherwise; $\ell > 0$ is a constant scalar of margin. Note we found that the commonly adopted balance constraints in max-margin clustering models [6] either are unnecessary or do not help in our framework. We will address this issue in specific models.

Clearly by absorbing the slack variables $\boldsymbol{\xi}$, the optimisation problem (4) is equivalent to

$$\inf_{q(\Theta)} \mathcal{L}(q(\Theta)) + 2c \sum_i \max\left(0, \max_{k:k\neq y_i} \mathbb{E}_{\Theta \sim q(\Theta)}[\zeta_{ik}]\right) \tag{5}$$

where $\zeta_{ik} := \ell \, \mathbb{I}(y_i \neq k) - (\boldsymbol{\eta}_{y_i} - \boldsymbol{\eta}_k)^T \mathbf{x}_i$. Exact solution to (5) is hard to compute. An alternative approach is to approximate the posterior by assuming independence between random variables, *e.g.* variational inference. However, this is usually slow and susceptible to local optimal. In order to obtain an analytic optimal distribution $q$ that facilitates efficient Bayesian inference, we resort to the technique of Gibbs classifier [17] which approximates (in fact, upper bounds due to the convexity of max function) the second term in (5) by an expected hinge loss, *i.e.*, moving the expectation out of the max. This leads to our final formulation of BMC:

$$\inf_{q(\Theta)} \mathcal{L}(q(\Theta)) + 2c \sum_i \mathbb{E}_{\Theta \sim q(\Theta)}\left[\max\left(0, \max_{k:k\neq y_i} \zeta_{ik}\right)\right]. \tag{6}$$

Problem (6) is still much more challenging than existing RegBayes models [17], which are restricted to supervised learning with two classes only. Specifically, BMC allows multiple clusters/classes in an unsupervised setting, and the latent cluster membership $y_i$ needs to be inferred. This complicates the model and brings challenges for posterior inference, as addressed below. In a nutshell, our inference algorithms rely on two key steps by exploring data augmentation techniques. First, in order to tackle the multi-class case, we introduce auxiliary variables $s_i := \arg\max_{k:k\neq y_i} \zeta_{ik}$. Applying standard derivations in calculus of variation [24] and augmenting the model with $\{s_i\}$, we obtain an analytic form of the optimal solution to (6) by augmenting $\Theta$ (refer to Appendix A for details):

$$q(\Theta, \{s_i\}) \propto p(\Theta|\mathbf{X}) \prod_i \exp(-2c \max(0, \zeta_{is_i})). \tag{7}$$

Second, since the max term in (7) obfuscates efficient sampling, we apply the augmentation technique introduced by [17], which showed that $q(\Theta, \{s_i\})$ is identical to the marginal distribution of the *augmented* post-data posterior

$$q(\Theta, \{s_i\}, \{\lambda_i\}) \propto p(\Theta|\mathbf{X}) \prod_i \tilde{\phi}_i(\lambda_i|\Theta), \tag{8}$$

where $\tilde{\phi}_i(\lambda_i|\Theta) := \lambda_i^{-\frac{1}{2}} \exp\left(\frac{-1}{2\lambda_i}(\lambda_i + c\zeta_{is_i})^2\right)$. Here $\lambda_i$ is an augmented variable for $\mathbf{x}_i$ that has an generalised inverse Gaussian distribution [25] given $\Theta$ and $\mathbf{x}_i$.

Note that our two steps of data augmentation are exact and incur no approximation. With the augmented variables $(\{s_i\}, \{\lambda_i\})$, we can develop efficient sampling algorithms for the augmented posterior $q(\Theta, \{s_i\}, \{\lambda_i\})$ without restrictive assumptions, thereby allowing us to approach the true target posterior $q(\Theta)$ by dropping the augmented variables. The details will become clear soon in our subsequent clustering models.

## 4  Dirichlet Process Max-margin Gaussian Mixture Models

In (4), we have left unspecified the prior $p(\Theta)$ and the likelihood $p(\mathbf{X}|\Theta)$. This section presents an instantiation of Bayesian nonparametric clustering for non-Gaussian data. We will present another instantiation of max-margin document clustering based on topic models in next section.

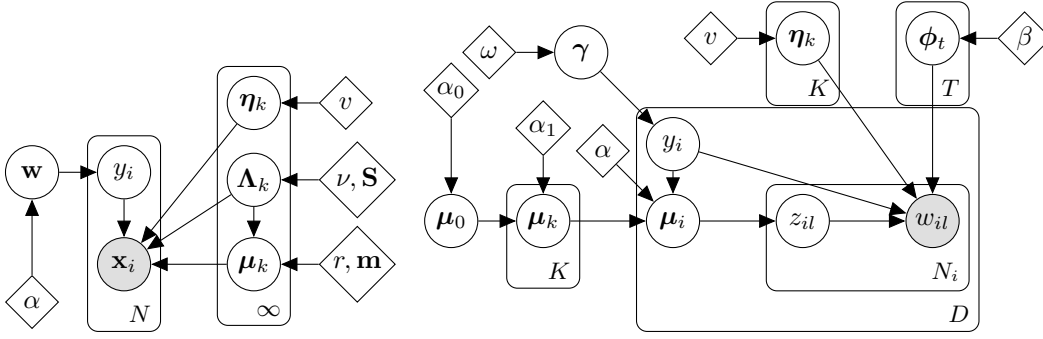

Figure 1: Left: Graphical model of DPMMGM. The part excluding $\boldsymbol{\eta}_k$ and $v$ corresponds to DPGMM. Right: Graphical model of MMCTM. The one excluding $\{\boldsymbol{\eta}_k\}$ and the arrow between $y_i$ and $w_{il}$ corresponds to CTM.

Here a convenient model of $p(\mathbf{X}, \Theta)$ is mixture of Gaussian. Let the mean and variance of the $k$-th cluster component be $\boldsymbol{\mu}_k$ and $\boldsymbol{\Lambda}_k$. In a nonparametric setting, the number of clusters is allowed to be infinite, and the cluster $y_i$ that each data point belongs to is drawn from a Dirichlet process [10]. To summarize, the latent variables are $\Theta = \{\boldsymbol{\mu}_k, \boldsymbol{\Lambda}_k, \boldsymbol{\eta}_k\}_{k=1}^{\infty} \cup \{y_i\}_{i=1}^{n}$. The prior $p(\Theta)$ is specified as: $\boldsymbol{\mu}_k$ and $\boldsymbol{\Lambda}_k$ employ a standard *Normal-inverse Wishart* prior [26]:

$$\boldsymbol{\mu}_k \sim \mathcal{N}(\boldsymbol{\mu}_k; \mathbf{m}, (r\boldsymbol{\Lambda}_k)^{-1}), \text{ and } \boldsymbol{\Lambda}_k \sim \mathcal{IW}(\boldsymbol{\Lambda}_k; \mathbf{S}, \nu). \tag{9}$$

$y_i \in \mathbb{Z}^+$ has a Dirichlet process prior with parameter $\alpha$. $\boldsymbol{\eta}_k$ follows a normal prior with mean $\mathbf{0}$ and variance $v\mathbf{I}$, where $\mathbf{I}$ is the identity matrix. The likelihood $p(\mathbf{x}_i|\Theta)$ is $\mathcal{N}(\mathbf{x}_i; \boldsymbol{\mu}_{y_i}, (r\boldsymbol{\Lambda}_{y_i})^{-1})$, *i.e.* independent of $\boldsymbol{\eta}_k$. The max-margin constraints take effects in the model via $\tilde{\phi}_i$'s in (8). Note this model of $p(\Theta, \mathbf{X})$, apart from $\boldsymbol{\eta}_k$, is effectively the Dirichlet process Gaussian mixture model [10] (DPGMM). Therefore, we call our post-data posterior $q(\Theta, \{s_i\}, \{\lambda_i\})$ in (8) as *Dirichlet process max-margin Gaussian mixture* model (DPMMGM). The hyperparameters include $\mathbf{m}, r, \mathbf{S}, \nu, \alpha, v$.

**Interpretation as a generalised DP mixture**  The formula of the augmented post-data posterior in (8) reveals that, compared with DPGMM, each data point is associated with an additional factor $\tilde{\phi}_i(\lambda_i|\Theta)$. Thus we can interpret DPMMGM as a generalised DP mixture with *Normal-inversed Wishart-Normal* as the base distribution, and a generalised *pseudo* likelihood that is proportional to

$$f(\mathbf{x}_i, \lambda_i | y_i, \boldsymbol{\mu}_{y_i}, \boldsymbol{\Lambda}_{y_i}, \{\boldsymbol{\eta}_k\}) := \mathcal{N}(\mathbf{x}_i; \boldsymbol{\mu}_{y_i}, (r\boldsymbol{\Lambda}_{y_i})^{-1}) \tilde{\phi}_i(\lambda_i|\Theta) . \tag{10}$$

To summarise, DPMMGM employs the following generative process with the graphical model shown in Fig. 1 (left):

$$(\boldsymbol{\mu}_k, \boldsymbol{\Lambda}_k, \boldsymbol{\eta}_k) \sim \mathcal{N}\left(\boldsymbol{\mu}_k; \mathbf{m}, (r\boldsymbol{\Lambda}_k)^{-1}\right) \times \mathcal{IW}\left(\boldsymbol{\Lambda}_k; \mathbf{S}, \nu\right) \times \mathcal{N}\left(\boldsymbol{\eta}_k; 0, v\mathbf{I}\right), \quad k = 1, 2, \cdots$$

$$\mathbf{w} \sim \text{Stick-Breaking}(\alpha),$$

$$y_i|\mathbf{w} \sim \text{Discrete}(\mathbf{w}), \qquad\qquad\qquad\qquad i \in [n]$$

$$(\mathbf{x}_i, \lambda_i)|y_i, \{\boldsymbol{\mu}_k, \boldsymbol{\Lambda}_k, \boldsymbol{\eta}_k\} \simeq f(\mathbf{x}_i, \lambda_i|y_i, \boldsymbol{\mu}_{y_i}, \boldsymbol{\Lambda}_{y_i}, \{\boldsymbol{\eta}_k\}). \qquad\qquad i \in [n]$$

Here $[n] := \{1, \cdots, n\}$ is the set of integers up to $n$ and $\simeq$ means that $(\mathbf{x}_i, \lambda_i)$ is generative from a distribution that is *proportional* to $f(\cdot)$. Since this normalisation constant is shared by all samples $\mathbf{x}_i$, there is no need to deal with it by posterior inference. Another benefit of this interpretation is that it allows us to use existing techniques for non-conjugate DP mixtures to sample the cluster indicators $y_i$ efficiently, and to infer the number of clusters in the data. This approach is different from previous work on RegBayes nonparametric models where truncated approximation is used to deal with the infinite dimensional model space [15, 18]. In contrast, our method does not rely on any approximation. Note that DPMMGM does not need the complicated class balance constraints [6] because the Gaussians in the pseudo likelihood would balance the clusters to some extent.

**Posterior inference**  Posterior inference for DPMMGM can be done by efficient Gibbs sampling. We integrate out the infinite dimension vector $\mathbf{w}$, so the variables needed to be sampled are $\{\boldsymbol{\mu}_k, \boldsymbol{\Lambda}_k, \boldsymbol{\eta}_k\}_k \cup \{y_i, s_i, \lambda_i\}_i$. Conditional distributions are derived in Appendix B. Note that we use an extension of the *Reused Algorithm* [27] to jointly sample $(y_i, s_i)$, which allows it to allocate to empty clusters in Bayesian nonparametric setting. The time complexity is almost the same as DPGMM except for the additional step to sample $\boldsymbol{\eta}_k$, with cost $O(p^3)$. So it would be necessary to put the constraints on a subspace (*e.g.*, by projection) of the original feature space when $p$ is high.

# 5 Max-margin Clustering Topic Model

Although many applications exhibit clustering structures in the raw observed data which can be effectively captured by DPMMGM, it is common that such regularities are more salient in terms of some high-level but latent features. For example, topic distributions are often more useful than word frequency in the task of document clustering. Therefore, we develop a *max-margin clustering topic model* (MMCTM) in the framework of BMC, which allows topic discovery to co-occur with document clustering in a Bayesian and max-margin fashion. To this end, the latent Dirichlet allocation (LDA) [28] needs to be extended by introducing a *cluster label* into the model, and define each cluster as a mixture of topic distributions. This cluster-based topic model [29] (CTM) can then be used in concert with BMC to enforce large margin between clusters in the posterior $q(\Theta)$.

Let $V$ be the size of the word vocabulary, $T$ be the number of topics, and $K$ be the number of clusters, $\mathbf{1}_N$ be a $N$-dimensional one vector. Then the generative process of CTM for the documents goes as:
1. For each topic $t$, generate its word distribution $\boldsymbol{\phi}_t$: $\boldsymbol{\phi}_t|\beta \sim \mathrm{Dir}(\beta\mathbf{1}_V)$.

2. Draw a base topic distribution $\boldsymbol{\mu}_0$: $\boldsymbol{\mu}_0|\alpha_0 \sim \mathrm{Dir}(\alpha_0\mathbf{1}_T)$. Then for each cluster $k$, generate its topic distribution mixture $\boldsymbol{\mu}_k$: $\boldsymbol{\mu}_k|\alpha_1, \boldsymbol{\mu}_0 \sim \mathrm{Dir}(\alpha_1\boldsymbol{\mu}_0)$.

3. Draw a base cluster distribution $\boldsymbol{\gamma}$: $\boldsymbol{\gamma}|\omega \sim \mathrm{Dir}(\omega\mathbf{1}_K)$. Then for each document $i \in [D]$:
   - Generate a *cluster label* $y_i$ and a *topic distribution* $\boldsymbol{\mu}_i$: $y_i|\boldsymbol{\gamma} \sim \mathrm{Discrete}(\boldsymbol{\gamma})$, $\boldsymbol{\mu}_i|\alpha, \boldsymbol{\mu}_{y_i} \sim \mathrm{Dir}(\alpha\boldsymbol{\mu}_{y_i})$.
   - Generate the observed words $w_{il}$: $z_{il} \sim \mathrm{Discrete}(\boldsymbol{\mu}_i)$, $w_{il} \sim \mathrm{Discrete}(\boldsymbol{\phi}_{z_{il}})$, $\forall\, l \in [N_i]$.

Fig. 1 (right) shows the structure. We then augment CTM with max-margin constraints, and get the same posterior as in Eq. (7), with the variables $\Theta$ corresponding to $\{\boldsymbol{\phi}_t\}_{t=1}^T \cup \{\boldsymbol{\eta}_k, \boldsymbol{\mu}_k\}_{k=1}^K \cup \{\boldsymbol{\mu_0}, \boldsymbol{\gamma}\} \cup \{\boldsymbol{\mu}_i, y_i\}_{i=1}^D \cup \{z_{il}\}_{i=1,l=1}^{D,N_i}$.

Compared with the raw word space which is normally extremely high-dimensional and sparse, it is more reasonable to characterise the clustering structure in the latent feature space–the empirical latent topic distributions as in the MedLDA [16]. Specifically, we summarise the topic distribution of document $i$ by $\mathbf{x}_i \in \mathbb{R}^T$, whose $t$-th element is $\frac{1}{N_i}\sum_{l=1}^{N_i}\mathbb{I}(z_{il}=t)$. Then the compatibility score for document $i$ with respect to cluster $k$ is defined similar to (3) as $F_k(\mathbf{x}_i) = \mathbb{E}_{q(\Theta)}\left[\boldsymbol{\eta}_k^T \mathbf{x}_i\right]$. Note, however, the expectation is also taken over $\mathbf{x}_i$ since it is *not* observed.

**Posterior inference**    To achieve fast mixing, we integrate out $\{\boldsymbol{\phi}_t\}_{t=1}^T \cup \{\boldsymbol{\mu}_0, \boldsymbol{\gamma}\} \cup \{\boldsymbol{\mu}_k\}_{k=1}^K \cup \{\boldsymbol{\mu}_i\}_{i=1}^D$ in the posterior, thus $\Theta = \{y_i\}_{i=1}^D \cup \{\boldsymbol{\eta}_k\}_{k=1}^K \cup \{z_{il}\}_{i=1,l=1}^{D,N_i}$. The integration is straightforward by the Dirichlet-Multinomial conjugacy. The detailed form of the posterior and the conditional distributions are derived in Appendix C. By extending CTM with max-margin, we note that many of the the sampling formulas are extension of those in CTM [29], with additional sampling for $\boldsymbol{\eta}_k$, thus the sampling can be done fairly efficiently.

**Dealing with vacuous solutions**    Different from DPMMGM, the max-margin constraints in MM-CTM do not interact with the observed words $w_{il}$, but with the latent topic representations $\mathbf{x}_i$ (or $z_{il}$) that are also inferred from the model. This easily makes the latent representation $z_i$'s collapse into a single cluster, a vacuous solution plaguing many other unsupervised learning methods as well. One remedy is to incorporate the cluster balance constraints into the model [7]. However, this does not help in our Bayesian setting because apart from significant increase in computational cost, MCMC often fails to converge in practice[2]. Another solution is to morph the problem into a weakly semi-supervised setting, where we assign to each cluster a few documents according to their true label (we will refer to these documents as *landmarks*), and sample the rest as in the above unsupervised setting. These "labeled examples" can be considered as introducing constraints that are alternative to the balance constraints. Usually only a very small number of labeled documents are needed, thus barely increasing the cost in training and labelling. We will focus on this setting in experiment.

# 6 Experiments

## 6.1 Dirichlet Process Max-margin Gaussian Mixture

We first show the distinction between our DPMMGM and DPGMM by running both models on the non-Gaussian *half-rings* data set [30]. There are a number of hyperparameters to be determined, *e.g.*, $(\alpha, r, S, \nu, v, c, \ell)$; see Section 4. It turns out the cluster structure is insensitive to $(\alpha, r, S, \nu)$, and so we use a standard sampling method to update $\alpha$ [31], while $r, \nu, S$ are sampled by employing Gamma, truncated Poisson, inverse Wishart priors respectively, as is done in [32]. We set $v = 0.01, c = 0.1, \ell = 5$ in this experiment. Note that the clustering structure is sensitive to the values of $c$ and $\ell$, which will be studied below. Empirically we find that DPMMGM converges much faster than DPGMM, both converging well within 200 iterations (see Appendix D.4 for examples). In Fig. 2, the clustering structures demonstrate clearly that DPMMGM relaxes the Gaussian assumption of the data distribution, and correctly finds the number of clusters based on the margin boundary, whereas DPGMM produces a too fragmented partition of the data for the clustering task.

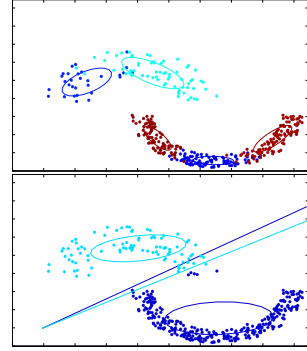

Figure 2: An illustration of DPGMM (up) and DPM-MGM (bottom).

**Parameter sensitivity** We next study the sensitivity of hyperparameters $c$ and $\ell$, with other hyperparameters sampled during inference as above. Intuitively the impact of these parameters is as follows. $c$ controls the weight that the max-margin constraint places on the posterior. If there were no other constraint, the max-margin constraint would drive the data points to collapse into a single cluster. As a result, we expect that a larger value of the weight $c$ will result in fewer clusters. Similarly, increasing the value of $\ell$ will lead to a higher loss for any violation of the constraints, thus driving the data points to collapse as well. To test these implications, we run DPMMGM on a 2-dimensional synthetic dataset with 15 clusters [33]. We vary $c$ and $\ell$ to study how the cluster structures change with respect to these parameter settings. As can be observed from Fig. 3, the results indeed follow our intuition, providing a mean to control the cluster structure in applications.

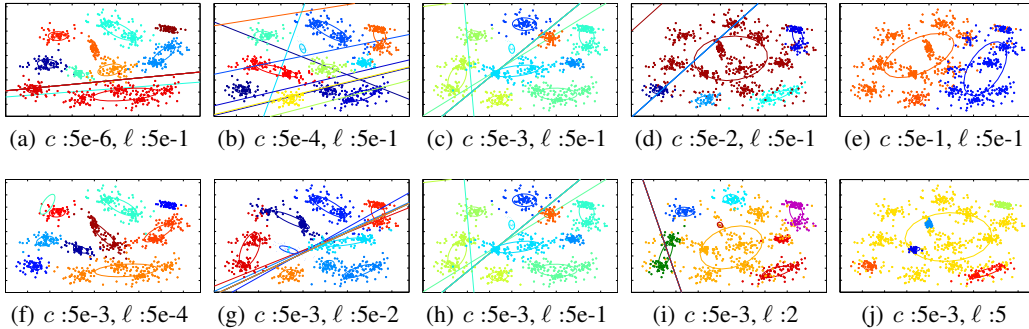

(a) $c$ :5e-6, $\ell$ :5e-1    (b) $c$ :5e-4, $\ell$ :5e-1    (c) $c$ :5e-3, $\ell$ :5e-1    (d) $c$ :5e-2, $\ell$ :5e-1    (e) $c$ :5e-1, $\ell$ :5e-1

(f) $c$ :5e-3, $\ell$ :5e-4    (g) $c$ :5e-3, $\ell$ :5e-2    (h) $c$ :5e-3, $\ell$ :5e-1    (i) $c$ :5e-3, $\ell$ :2    (j) $c$ :5e-3, $\ell$ :5

Figure 3: Clustering structures with varied $\ell$ and $c$: (first row) fixed $\ell$ and increasing $c$; (second row) fixed $c$ and increasing $\ell$. Lines are $\eta$'s. Clearly the number cluster decreases with growing $c$ and $\ell$.

**Real Datasets**. As other clustering models, we test DPMMGM on ten real datasets (small to moderate sizes) from the UCI repository [34]. Scaling up to large dataset is an interesting future. The first three columns of Table 1 list some of the statistics of these datasets (we used random subsets of the three large datasets – Letter, MNIST, and Segmentation).

**A heuristic approach for model selection**. Model selection is generally hard for unsupervised clustering. Most existing algorithms simply fix the hyperparameters without examining their impacts on model performance [10, 35]. In DPMMGM, the hyperparameters $c$ and $\ell$ are critical to clustering quality since they control the number of clusters. Without training data in our setting they can not be set using cross validation. Moreover, they are not feasible to be estimated use Bayesian sampling as well because they are not parameters from a proper Bayesian model. we thus introduce a time-efficient heuristic approach to selecting appropriate values. Suppose the dataset is known to have $K$ clusters. Our heuristic goes as follows. First initialise $c$ and $\ell$ to 0.1. Then at each iteration, we compare the inferred number of clusters with $K$. If it is larger than $K$ (otherwise we do the converse), we choose $c$ or $\ell$ randomly, and increase its value by $\frac{u}{n}$, where $u$ is a uniform random variable in $[0, 1]$ and $n$ is the number of iterations so far. According to the *parameter sensitivity* studied above, increasing $c$ or $\ell$ tends to decrease the number of clusters, and the model eventually

| Dataset | Data property | | | NMI | | | | |
|---|---|---|---|---|---|---|---|---|
| | $n$ | $p$ | $K$ | kmeans | nCut | DPGMM | DPMMGM | DPMMGM* |
| Glass | 214 | 10 | 7 | $0.37_{\pm 0.04}$ | $0.22_{\pm 0.00}$ | $0.37_{\pm 0.05}$ | $\mathbf{0.46}_{\pm 0.01}$ | $0.45_{\pm 0.01}$ |
| Half_circle | 300 | 2 | 2 | $0.43_{\pm 0.00}$ | $1.00_{\pm 0.00}$ | $0.49_{\pm 0.02}$ | $0.67_{\pm 0.02}$ | $0.51_{\pm 0.07}$ |
| Iris | 150 | 4 | 3 | $0.72_{\pm 0.08}$ | $0.61_{\pm 0.00}$ | $0.73_{\pm 0.00}$ | $\mathbf{0.73}_{\pm 0.00}$ | $0.73_{\pm 0.00}$ |
| Letter | 1000 | 16 | 10 | $0.33_{\pm 0.01}$ | $0.04_{\pm 0.00}$ | $0.19_{\pm 0.09}$ | $\mathbf{0.38}_{\pm 0.04}$ | $0.23_{\pm 0.04}$ |
| MNIST | 1000 | 784 | 10 | $0.50_{\pm 0.01}$ | $0.38_{\pm 0.00}$ | $0.55_{\pm 0.03}$ | $\mathbf{0.56}_{\pm 0.01}$ | $0.55_{\pm 0.02}$ |
| Satimage | 4435 | 36 | 6 | $0.57_{\pm 0.06}$ | $0.55_{\pm 0.00}$ | $0.21_{\pm 0.05}$ | $0.51_{\pm 0.01}$ | $0.30_{\pm 0.00}$ |
| Segment'n | 1000 | 19 | 7 | $0.52_{\pm 0.03}$ | $0.34_{\pm 0.00}$ | $0.23_{\pm 0.09}$ | $\mathbf{0.61}_{\pm 0.05}$ | $0.52_{\pm 0.10}$ |
| Vehicle | 846 | 18 | 4 | $0.10_{\pm 0.00}$ | $0.14_{\pm 0.00}$ | $0.02_{\pm 0.02}$ | $\mathbf{0.14}_{\pm 0.00}$ | $0.05_{\pm 0.00}$ |
| Vowel | 990 | 10 | 11 | $0.42_{\pm 0.01}$ | $0.44_{\pm 0.00}$ | $0.28_{\pm 0.03}$ | $0.39_{\pm 0.02}$ | $0.41_{\pm 0.02}$ |
| Wine | 178 | 13 | 3 | $0.84_{\pm 0.01}$ | $0.46_{\pm 0.00}$ | $0.56_{\pm 0.02}$ | $\mathbf{0.90}_{\pm 0.02}$ | $0.59_{\pm 0.01}$ |

Table 1: Comparison for different methods on NMI scores. $K$: true number of clusters.

stabilises due to the stochastic decrement by $\frac{u}{n}$. We denote the model learned from this heuristic as DPMMGM. In the case where the true number of clusters is unknown, we can still apply this strategy, except that the number of clusters $K$ needs to be first inferred from DPGMM. This method is denoted as DPMMGM*.

**Comparison**. We measure the quality of clustering results by using the standard normalised mutual information (NMI) criterion [36]. We compare our DPMMGM with the well established KMeans, nCut and DPGMM clustering methods[3]. All experiments are repeated for five times with random initialisation. The results are shown in Table 1. Clearly DPMMGM significantly outperforms other models, achieving the best NMI scores. DPMMGM*, which is not informed of the true number of clusters, still obtains reasonably high NMI scores, and outperforms the DPGMM model.

## 6.2 Max-margin Clustering Topic Model

**Datasets**. We test the MMCTM model on two document datasets: *20NEWS* and *Reuters-R8* . For the *20NEWS* dataset, we combine the training and test datasets used in [16], which ends up with 20 categories/clusters with roughly balanced cluster sizes. It contains 18,772 documents in total with a vocabulary size of 61,188. The *Reuters-R8* dataset is a subset of the Reuters-21578 dataset[4], with of 8 categories and 7,674 documents in total. The size of different categories is biased, with the lowest number of documents in a category being 51 while the highest being 2,292.

**Comparison** We choose $L \in \{5, 10, 15, 20, 25\}$ documents randomly from each category as the *landmarks*, use 80% documents for training and the rest for testing. We set the number of topics (i.e., $T$) to 50, and set the Dirichlet prior in Section 5 to $\omega = 0.1$, $\beta = 0.01$, $\alpha = \alpha_0 = \alpha_1 = 10$, as clustering quality is not sensitive to them. For the other hyperparameters related to the max-margin constraints, *e.g.*, $v$ in the Gaussian prior for $\boldsymbol{\eta}$, the balance parameter $c$, and the cost parameter $\ell$, instead of doing cross validation which is computationally expensive and not helpful for our scenario with few labeled data, we simply set $v = 0.1, c = 9, \ell = 0.1$. This is found to be a good setting and denoted as MMCTM. To test the robustness of this setting, we vary $c$ over $\{0.1, 0.2, 0.5, 0.7, 1, 3, 5, 7, 9, 15, 30, 50\}$ and keep $v = \ell = 0.1$ ($\ell$ and $c$ play similar roles and so varying one is enough). We choose the best performance out of these parameter settings, denoted as MMCTM*, which can be roughly deemed as the setting for the optimal performance. We compared MMCTM with state-of-the-art SVM and semi-supervised SVM (S3VM) models. They are efficiently implemented in [37], and the related parameters are chosen by 5-fold cross validation. As in [16], raw word frequencies are used as input features. We also compare MMCTM with a Bayesian baseline–cluster based topic model (CTM) [29], the building block of MMCTM without the max-margin constraints. Note we did not compare with the standard MedLDA [16] because it is supervised. We measure the performance by cluster accuracy, which is the proportion of correctly clustered documents. To accelerate MMCTM, we simply initialise it with CTM, and find it converges surprisingly fast in term of accuracy, *e.g.*, usually within 30 iterations (refer to Appendix

| L | CTM | SVM | S3VM | MMCTM | MMCTM* |
|---|-----|-----|------|-------|--------|
| *20NEWS* | | | | | |
| 5 | $17.22_{\pm 4.2}$ | $37.13_{\pm 2.9}$ | $39.36_{\pm 3.2}$ | $\mathbf{56.70}_{\pm 1.9}$ | $\mathbf{57.86}_{\pm 0.9}$ |
| 10 | $24.50_{\pm 4.5}$ | $46.99_{\pm 2.4}$ | $47.91_{\pm 2.8}$ | $54.92_{\pm 1.6}$ | $\mathbf{56.56}_{\pm 1.3}$ |
| 15 | $22.76_{\pm 4.2}$ | $52.80_{\pm 1.2}$ | $52.49_{\pm 1.4}$ | $55.06_{\pm 2.7}$ | $\mathbf{57.80}_{\pm 2.2}$ |
| 20 | $26.07_{\pm 7.2}$ | $56.10_{\pm 1.5}$ | $54.44_{\pm 2.1}$ | $56.62_{\pm 2.2}$ | $\mathbf{59.70}_{\pm 1.4}$ |
| 25 | $27.20_{\pm 1.5}$ | $59.15_{\pm 1.4}$ | $57.45_{\pm 1.7}$ | $55.70_{\pm 2.4}$ | $\mathbf{61.92}_{\pm 3.0}$ |
| *Reuters-R8* | | | | | |
| 5 | $41.27_{\pm 16.7}$ | $78.12_{\pm 1.1}$ | $78.51_{\pm 2.3}$ | $79.18_{\pm 4.1}$ | $80.86_{\pm 2.9}$ |
| 10 | $42.63_{\pm 7.4}$ | $80.69_{\pm 1.2}$ | $79.15_{\pm 1.2}$ | $80.04_{\pm 5.3}$ | $\mathbf{83.48}_{\pm 1.0}$ |
| 15 | $39.67_{\pm 9.9}$ | $83.25_{\pm 1.7}$ | $81.87_{\pm 0.8}$ | $85.48_{\pm 2.1}$ | $86.86_{\pm 2.5}$ |
| 20 | $58.24_{\pm 8.3}$ | $85.66_{\pm 1.0}$ | $73.95_{\pm 2.0}$ | $82.92_{\pm 1.7}$ | $83.82_{\pm 1.6}$ |
| 25 | $51.93_{\pm 5.9}$ | $84.95_{\pm 0.1}$ | $82.39_{\pm 1.8}$ | $86.56_{\pm 2.5}$ | $\mathbf{88.12}_{\pm 0.5}$ |

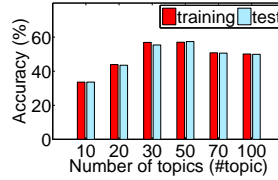

(a) *20NEWS* dataset

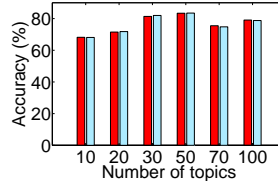

(b) *Reuters-R8* dataset

Table 2: Clustering acc. (in %). **Bold** means significantly different.    Figure 4: Accuracy vs. #topic

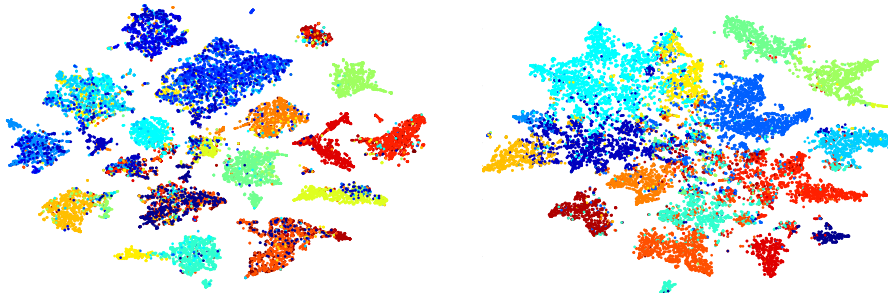

Figure 5: 2-D tSNE embedding on *20NEWS* for MMCTM (left) and CTM (right). Best viewed in color. See Appendix D.3 for the results on *Reuters-R8* datasets.

D.5). The accuracies are shown in Table 2, and we can see that MMCTM outperforms other models (also see Appendix D.4), except for SVM when $L = 20$ on the Reuters-R8 dataset. In addition, MMCTM performs almost as well as using the optimal parameter setting (MMCTM*).

**Sensitivity to the number of topics (i.e., $T$).** Note the above experiments simply set $T = 50$. To validate the affect of $T$, we varied $T$ from 10 to 100, and the corresponding accuracies are plotted In Fig. 4 for the two datasets. In both cases, $T = 50$ seems to be a good parameter value.

**Cluster embedding.**    We finally plot the clustering results by embedding them into the 2-dimensional plane using tSNE [38]. In Fig. 5, it can be observed that compared to CTM, MMCTM generates well separated clusters with much larger margin between clusters.

## 7    Conclusions

We propose a robust Bayesian max-margin clustering framework to bridge the gap between max-margin learning and Bayesian clustering, allowing many Bayesian clustering algorithms to be directly equipped with the max-margin criterion. Posterior inference is done via two data augmentation techniques. Two models from the framework are proposed for Bayesian nonparametric max-margin clustering and topic model based document clustering. Experimental results show our models significantly outperform existing methods with competitive clustering accuracy.

**Acknowledgments**

This work was supported by an Australia China Science and Research Fund grant (ACSRF-06283) from the Department of Industry, Innovation, Climate Change, Science, Research and Tertiary Education of the Australian Government, the National Key Project for Basic Research of China (No. 2013CB329403), and NSF of China (Nos. 61322308, 61332007). NICTA is funded by the Australian Government as represented by the Department of Broadband, Communications and the Digital Economy and the Australian Research Council through the ICT Centre of Excellence program.

## Footnotes

[1]In theory, we also require that $q$ is absolutely continuous with respect to $p$ to make the KL-divergence well defined. The present paper treats this constraint as an implicit assumption for clarity.

[2]We observed the cluster sizes kept bouncing with sampling iterations. This is probably due to the highly nonlinear mapping from observed word space to the feature space (topic distribution), making the problem multi-modal, *i.e.*, there are multiple optimal topic assignments in the post-data posterior (8). Also the balance constraints might weaken the max-margin constraints too much.

[3]We additionally show some comparison with some existing max-margin clustering models in Appendix D.2 on two-cluster data because their code only deals with the case of two clusters. Our method performs best.

[4]Downloaded from csmining.org/index.php/r52-and-r8-of-reuters-21578.html.

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
