[Supplementary Material]

# A  Derivation of Eq.(7)

First we introduce a basic lemma from calculus of variation [24] below:

**Lemma 1 (The Euler-Lagrange equation)** *Let $F(\alpha, \beta, \gamma)$ be a function with continuously first and second partial derivatives with respect to $(\alpha, \beta, \gamma)$, $\mathcal{S} = \{x \in \mathcal{C}^1[a,b] | x(a) = y_a, x(b) = y_b\}$ where $\mathcal{C}^1[a,b]$ denotes the space of one time continuously differentiable functions in $[a,b]$. Let $\mathcal{I} : \mathcal{S} \to \mathbb{R}$ be a function of the form:*

$$\mathcal{I}(x) = \int_a^b F(x(t), x'(t), t) \mathrm{d}t .$$

*If $\mathcal{I}$ has an extremum at $x_0 \in \mathcal{S}$, then $x_0$ satisfies the Euler-Lagrange equation:*

$$\frac{\partial F}{\partial \alpha}(x_0(t), x'_0(t), t) - \frac{\mathrm{d}}{\mathrm{d}t}\left(\frac{\partial F}{\partial \beta}(x_0(t), x'_0(t), t)\right) = 0,$$

*for $t \in [a,b]$.*

**Derivation of Eq.(7)**  Substituting the $\mathcal{L}$ into (6), it is easy to see that (6) is equivalent to

$$\inf_{q(\Theta)} \mathrm{KL}(q(\Theta)||p(\Theta)) - \mathbb{E}_{\Theta \sim q(\Theta)}\left[\log p(\mathbf{X}|\Theta)\right] + 2c\sum_i \mathbb{E}_{\Theta \sim q(\Theta)}\left[\max\left(0, \max_{k:k \neq y_i} \zeta_{ik}\right)\right]$$

$$\implies \inf_{q(\Theta)} \int_\Theta \log \frac{q(\Theta)}{p(\Theta)} q(\Theta) \mathrm{d}\Theta - \int_\Theta \log p(\mathbf{X}|\Theta) q(\Theta) \mathrm{d}\Theta + 2c\sum_i \int_\Theta \left[\max\left(0, \max_{k:k \neq y_i} \zeta_{ik}\right)\right] q(\Theta) \mathrm{d}\Theta$$

Let $F$ in Lemma 1 be:

$$F(q(\Theta), q'(\Theta), \Theta) = \log \frac{q(\Theta)}{p(\Theta)} q(\Theta) - \log p(\mathbf{X}|\Theta) q(\Theta) + 2c\sum_i \left[\max\left(0, \max_{k:k \neq y_i} \zeta_{ik}\right)\right] q(\Theta) .$$

Applying Lemma 1 and simplifying we get

$$q(\Theta) \propto p(\Theta|\mathbf{X}) \prod_i \exp(-2c \max(0, \max_{k:k \neq y_i} \zeta_{ik})) .$$

Now with similar idea as the slice sampler [39], we augment the state $\Theta$ with some auxiliary variables $(s_i \in [K])_{i=1}^n$ such that the conditional probability of $s_i$ is:

$$p(s_i = s|\text{others}) = \delta_{\arg\max_{k:k \neq y_i} \zeta_{ik}}(s) ,$$

where $\delta_x()$ is a spike at location $x$. Combining the above probability, the augmented posterior is then given by

$$q(\Theta, \{s_i\}) \propto p(\Theta|\mathbf{X}) \prod_i \exp(-2c \max(0, \zeta_{is_i})) ,$$

which is exactly Eq.(7). Note this augmentation allows us to sample $\Theta$ and $\{s_i\}$ via Gibbs sampler:

- Conditioned on $\{s_i\}$, sample $\Theta$. This results in simple conditional distributions similar to the 2-class case.

- Conditioned on $\Theta$, update $\{s_i\}$ (possibly jointly sample $(y_i, s_i)$ as is done in this paper), with $s_i$ simply takes the value: $s_i = \arg\max_{k:k \neq y_i} \zeta_{ik}$.

# B  Posterior formulas for the DPMMGM model

The posterior of the DPMMGM model is obtained by multiplying the pseudo likelihood and the prior in the generative process in Section 4. We use MCMC for posterior inference since it has been shown quite efficient [17]. After integrating out the mixing weights $\mathbf{w}$, the variables that need to be sampled are $C := \Theta \cup \{s_i, \lambda_i\}_i = \{\boldsymbol{\mu}_k, \boldsymbol{\Lambda}_k, \boldsymbol{\eta}_k\}_k \cup \{y_i, s_i, \lambda_i\}_i$. Substituting the DPGMM posterior into (8) we can derive the following updates for the variables.

**Sampling $\boldsymbol{\eta}_k$**   This can be done by expanding related $\tilde{\phi}_i(\lambda_i|\Theta)$ terms in (8). Clearly the terms associated with $\boldsymbol{\eta}_k$ are those $i$'s such that $y_i = k$ or $s_i = k$. First expanding the squared term in $\tilde{\phi}_i(\lambda_i|\Theta)$, we have

$$(\lambda_i + c\zeta_{is_i})^2 = \left(\lambda_i + c\ell - c\left(\boldsymbol{\eta}_{y_i} - \boldsymbol{\eta}_{s_i}\right)^T \mathbf{x}_i\right)^2$$

$$= (\lambda_i + c\ell)^2 - 2c\left(\lambda_i + c\ell\right)\left(\boldsymbol{\eta}_{y_i} - \boldsymbol{\eta}_{s_i}\right)^T \mathbf{x}_i + c^2\boldsymbol{\eta}_{y_i}^T \mathbf{x}_i\mathbf{x}_i^T \boldsymbol{\eta}_{y_i} + c^2\boldsymbol{\eta}_{s_i}^T \mathbf{x}_i\mathbf{x}_i^T \boldsymbol{\eta}_{s_i} - 2c^2\boldsymbol{\eta}_{y_i}^T \mathbf{x}_i\mathbf{x}_i^T \boldsymbol{\eta}_{s_i}$$

Now sum over all $i$'s such that $y_i = k$ or $s_i = k$ and multiply by $-\frac{1}{2\lambda_i}$ inside $\tilde{\phi}_i(\lambda_i|\Theta)$, we have the terms inside the exponent being:

$$\Longrightarrow \left(\sum_{i:y_i=k}\left(\frac{c}{\lambda_i}\left(\lambda_i + c\ell\right)\mathbf{x}_i^T + \frac{c^2}{\lambda_i}\boldsymbol{\eta}_{s_i}^T\mathbf{x}_i\mathbf{x}_i^T\right) - \left(\sum_{i:s_i=k}\left(\frac{c}{\lambda_i}\left(\lambda_i + c\ell\right)\mathbf{x}_i^T - \frac{c^2}{\lambda_i}\boldsymbol{\eta}_{y_i}^T\mathbf{x}_i\mathbf{x}_i^T\right)\right)\right)\boldsymbol{\eta}_k$$

$$-\frac{c^2}{2}\boldsymbol{\eta}_k^T\left(\sum_{i:y_i=k||s_i=k}\frac{\mathbf{x}_i\mathbf{x}_i^T}{\lambda_i}\right)\boldsymbol{\eta}_k \ .$$

This corresponds to the first order and second order terms of a Multivariate normal distribution. Re-arranging terms it is easy to see that $\boldsymbol{\eta}_k$ is a multivariate normal distribution as:

$q(\boldsymbol{\eta}_k|C\backslash\boldsymbol{\eta}_k) \sim \mathcal{N}(\boldsymbol{\eta}_k; \mathbf{B}^{-1}\mathbf{A}, \mathbf{B}^{-1})$, where

$$\mathbf{A} = c^2\left(\left(\sum_{i:y_i=k}(\ell\kappa_i + \frac{1}{c})\mathbf{x}_i\right) - \left(\sum_{i:s_i=k}(\ell\kappa_i + \frac{1}{c})\mathbf{x}_i\right) + \sum_{i:y_i=k}\kappa_i\mathbf{x}_i\mathbf{x}_i^T\boldsymbol{\eta}_{s_i} + \sum_{i:s_i=k}\kappa_i\mathbf{x}_i\mathbf{x}_i^T\boldsymbol{\eta}_{y_i}\right)$$

$$\mathbf{B} = \frac{1}{v}\mathbf{I} + c^2\left(\sum_{i:y_i=k \text{ or } s_i=k}\kappa_i\mathbf{x}_i\mathbf{x}_i^T\right), \tag{11}$$

with $\kappa_i := \lambda_i^{-1}$ for notation simplicity, and $K$ is the current number of active clusters.

**Sample $\lambda_i$**   It is easy to see that $\lambda_i$ has posterior

$$q(\lambda_i|C\backslash\lambda_i) \sim \mathcal{GIG}\left(\lambda_i; -\frac{1}{2}, 1, c^2\left(\ell - (\boldsymbol{\eta}_{y_i} - \boldsymbol{\eta}_{s_i})^T \mathbf{x}_i\right)^2\right),$$

where $\mathcal{GIG}(x; p, a, b) \propto x^{p-1}\exp\left(-\frac{1}{2}\left(\frac{b}{x} + ax\right)\right)$ is a generalised inverse Gaussian distribution with parameters $(p, a, b)$. So $\lambda_i$ can be sampled efficiently via the technique proposed recently by [40].

**Sample $(\boldsymbol{\mu}_k, \boldsymbol{\Lambda}_k)$**   With a *Gauss-inverse Wishart* prior for $(\boldsymbol{\mu}_k, \boldsymbol{\Lambda}_k)$ in (9), the posterior is still a *Gauss-inverse Wishart* [26]. Let $\bar{\mathbf{x}}_k = \frac{1}{n_k}\sum_{i:y_i=k}\mathbf{x}_i$ where $n_k = \sum_i \mathbb{I}(y_i = k)$ is the current number of observations in cluster $k$. Then the update formulae of parameters are

$$r_k = r + n_k, \ \ \nu_k = \nu + n_k, \ \ \mathbf{m}_k = \frac{r\mathbf{m} + n_k\bar{\mathbf{x}}_k}{r + n_k},$$

$$\mathbf{S}_k = \mathbf{S} + \frac{r \cdot n_k}{r + n_k}(\mathbf{m} - \bar{\mathbf{x}}_k)(\mathbf{m} - \bar{\mathbf{x}}_k)^T + \sum_{i:y_i=k}(\mathbf{x}_i - \bar{\mathbf{x}}_k)(\mathbf{x}_i - \bar{\mathbf{x}}_k)^T.$$

**Sample $(y_i, s_i)$ jointly with w integrated out**   Our model is nonparametric Bayesian, allowing new components to be born while sampling $y_i$, as is done in DPGMM. We sample $y_i$ in a collapsed fashion by integrating out the weights $\mathbf{w}$, which has been shown more efficient [41]. Because of the non-conjugacy between the pseudo likelihood and the prior, we adopt an auxiliary state MCMC algorithm called *Reused Algorithm*, developed by [27] recently. The idea is similar to *Algorithm 8* of [41] by introducing some augmented empty components into the MCMC states, but the efficiency is much improved. Since $s_i$ depends on $y_i$, we will sample $(y_i, s_i)$ jointly. The algorithm is summarised in Algorithm 1.

| Algorithm 1: Reused Algorithm for sampling $y_i$ |
| --- |

1: Initialise $\{y_1, \cdots, y_n\}$. Assign $H$ (set to 3 in the experiments) auxiliary empty clusters $(\boldsymbol{\mu}_t, \boldsymbol{\Lambda}_t, \boldsymbol{\eta}_t)_{t=1}^T$.
2: Remove $y_i$ from the statistics (denote the current cluster parameters as $(\boldsymbol{\mu}_k, \boldsymbol{\Lambda}_k, \boldsymbol{\eta}_k)$):
3: **if** the current cluster contains only $\mathbf{x}_i$ **then**
4:      Randomly sample a cluster from the $H$ auxiliary empty clusters;
5:      Assign the selected auxiliary empty clusters with parameters $(\boldsymbol{\mu}_k, \boldsymbol{\Lambda}_k, \boldsymbol{\eta}_k)$;
6: **end if**
7: Sample $(y_i, s_i)$ jointly from its posterior distribution:
$$q(y_i = k, s_i = \tau(k)|C\backslash(y_i, s_i)) \propto \beta_k f(\mathbf{x}_i|\cdots),$$
     where $\tau(k) = \arg\max_{k':k'\neq k} \zeta_{ik'}$, $f(\cdot)$ is defined in (10), $\beta_k = n_k$ if $n_k > 0$, and $\frac{\alpha}{H}$ otherwise.
8: **if** $y_i$ is assigned to an auxiliary empty cluster **then**
9:      Move the cluster out of the auxiliary empty cluster.
10:      Draw the cluster parameters for the corresponding auxiliary empty cluster from the prior distribution.
11: **end if**

## C   Posterior formulas for the MMCTM model

We first derive the posterior for the CTM (the one without maxi-margin constraint). This corresponds to Figure 1 (right) with $\{\boldsymbol{\eta}_k\}$ and the arrow between $y_i$ and $w_{i\ell}$ removed. We can write down the posterior of CTM by first integrating out the $\boldsymbol{\gamma}$ and $\{\boldsymbol{\phi}_t\}$ latent variables using the Dirichlet-Multinomial conjugacy. The hierarchical Dirichlet distribution part does not embody a close form posterior, but we can represent the posterior in a sequential way, *e.g.*, we assume the words are generated sequentially, then we multiply the probabilities of generating each word to form the posterior. Before giving the posterior formula, we first define the following statistics:

- $m_{tw}$: the number of times that a word $w$ is sampled from topic $t$.
- $C_k$: the number of documents in cluster $k$.
- $n_{it}$: the number of words in document $i$ with topic $t$.
- $g_{kt}$: the number of documents in cluster $k$ with topic $t$.
- $h_t$: the number of clusters included in topic $t$.

We further use 'dot' to represent marginal sum, *e.g.*, $n_{i\cdot} := \sum_{t=1}^T n_{it}$. Superscript "$\neg l$" means the statistics excluding word $l$, *e.g.*, $\mathbf{Z}^{\neg l} := \{z_{il'}\}_{i,l'\neq l}$. Based on these statistics, the posterior of CTM can be written as

$$p(\mathbf{W}, \{m_{tw}\}, \{C_k\}, \{n_{it}\}, \{g_{yt}\}, \{h_t\}|\alpha_0, \alpha_1, \alpha, \omega, \nu, \beta)$$
$$= \frac{\prod_{t=1}^T \Gamma(m_{t\cdot} + \beta)}{\Gamma(m_{\cdot\cdot} + \beta V)} \frac{\prod_{k=1}^K \Gamma(C_k + \omega)}{\Gamma(C_\cdot + \omega K)} \prod_{i=1}^D \prod_{l=1}^{N_d} p(z_{il}|y_i, \text{others}) , \tag{12}$$

where

$$p(z_{il}|y_i, \text{others}) = \frac{n_{iz_{il}}^< + \alpha \dfrac{g_{y_i z_{il}}^< + \alpha_1 \frac{h_{z_{il}}^< + \alpha_0/T}{h_\cdot^< + \alpha_0}}{g_{y_i\cdot}^< + \alpha_1}}{n_{i\cdot}^< + \alpha} .$$

Here superscript "$<$" means the statistics up to the current word in the current document, and $p(z_{il}|y_i, \text{others})$ represents the probability of generating the word $w_{il}$.

According to the Bayesian max-margin clustering framework in Section 3, the posterior of the proposed MMCTM is simply multiplying the above CTM posterior (12) with the terms $\tilde{\phi}_i(\lambda_i|\Theta)$ defined in (8) induced by the max-margin constraints, and with the features $\mathbf{x}_i$ defined as latent: $\mathbf{x}_i \triangleq \bar{\mathbf{z}}_i \in \mathbb{R}^T$, where the $t$-th element is simply $\frac{1}{N_i} \sum_{l=1}^{N_i} \mathbb{I}(z_{il} = t)$.

## C.1 Posterior inference

The variables needed to be sampled are $C = (\{\boldsymbol{\eta}_k\}, \{\lambda_i\}, \{z_{il}\}, \{y_i, s_i\})$, the conditional distributions can be derived from the posterior $q(\{\boldsymbol{\eta}_k\}, \{\lambda_i\}, \{z_{il}\}, \{y_i, s_i\})$. Based on the posterior analysis above, the sampling proceeds simply goes as:

**Sampling $\boldsymbol{\eta}_k$:** This is similar to the DPMMGM case since the original posterior $p(\mathbf{W}, \{m_{tw}\}, \{C_k\}, \{n_{it}\}, \{g_{kt}\}, \{h_t\} | \alpha_0, \alpha_1, \alpha, \omega, \nu, \beta)$ does not impact the $\boldsymbol{\eta}_k$ directly. So the form of the conditional distribution for $\boldsymbol{\eta}_k$ is the same as the DPMMGM (but note here $\mathbf{x}_i$ is latent):

$$q(\boldsymbol{\eta}_k | C \backslash \boldsymbol{\eta}_k) \sim \mathcal{N}(\boldsymbol{\eta}_k; \mathbf{B}^{-1}\mathbf{A}, \mathbf{B}^{-1}), \text{ where}$$

$$\mathbf{A} = c^2 \left( \left( \sum_{i:y_i=k} \frac{c\ell + \lambda_i}{c\lambda_i} \mathbf{x}_i \right) - \left( \sum_{i:s_i=k} \frac{c\ell + \lambda_i}{c\lambda_i} \mathbf{x}_i \right) \right.$$

$$\left. + \sum_{i:y_i=k} \frac{\mathbf{x}_i \mathbf{x}_i^T}{\lambda_i} \boldsymbol{\eta}_{s_i} + \sum_{i:s_i=k} \frac{\mathbf{x}_i \mathbf{x}_i^T}{\lambda_i} \boldsymbol{\eta}_{y_i} \right)$$

$$\mathbf{B} = \frac{1}{v}\mathbf{I} + c^2 \left( \sum_{i:y_i=k||s_i=k} \frac{\mathbf{x}_i \mathbf{x}_i^T}{\lambda_i} \right),$$

and $K$ is the current number of active clusters.

**Sampling $\lambda_i$:** This is also similar to the DPMMGM case. By taking related terms out of $\tilde{\phi}_i(\lambda_i | \Theta)$ induced by the max-margin constraint, we get the posterior as

$$q(\lambda_i | C \backslash \lambda_i) \sim \mathcal{GIG}\left( \lambda_i; -\frac{1}{2}, 1, c^2 \left( \ell - (\boldsymbol{\eta}_{y_i} - \boldsymbol{\eta}_{s_i})^T \mathbf{x}_i \right)^2 \right).$$

**Sampling $z_{il}$:** When sampling $z_{il}$, we first remove the word $l$ from document $i$ out of the statistics, and then add it in topic $t = (1, 2, \cdots, T)$. This would change the statistics, and so as the latent feature $\mathbf{x}_i$ of document $i$ in the max-margin term $\tilde{\phi}_i(\lambda_i | \Theta)$. By careful simplification, we get the resulting posterior as:

$$q(z_{il} = t | w_{il} = w, \text{others})$$

$$\propto p\left(z_{il} | y_i, \mathbf{Z}^{\neg l}, \text{others}\right) \frac{(m_{tw}^{\neg l} + \beta)}{m_{t\cdot}^{\neg l} + \beta V} \exp\left( \frac{A_t}{N_i} + \frac{B_{tt} + 2\sum_{t'=1}^{T} B_{tt'} n_{it'}^{\neg l}}{N_i^2} \right), \quad (13)$$

where

$$\mathbf{A} = \left( c + \frac{c^2 \ell}{\lambda_i} \right) (\boldsymbol{\eta}_{y_i} - \boldsymbol{\eta}_{s_i}),$$

$$\mathbf{B} = -\frac{c^2}{2\lambda_i} (\boldsymbol{\eta}_{y_i} - \boldsymbol{\eta}_{s_i})(\boldsymbol{\eta}_{y_i} - \boldsymbol{\eta}_{s_i})^T,$$

$$p\left(z_{il} | y_i, \mathbf{Z}^{\neg l}, \text{others}\right) = \frac{n_{iz_{il}}^{\neg l} + \alpha \frac{g_{y_i z_{il}}^{\neg l} + \alpha_1 \frac{h_{z_{il}}^{\neg l} + \alpha_0/T}{h_\cdot^{\neg l} + \alpha_0}}{g_{y_i\cdot}^{\neg l} + \alpha_1}}{n_{i\cdot}^{\neg l} + \alpha}.$$

Here the first and second terms in (13) come from (12) by variable elimination, the third term comes from the $\tilde{\phi}_i(\lambda_i | \Theta)$ terms induced by the max-margin constraints.

**Sampling $(y_i, s_i)$:** Because $y_i$ relates to all the words in document $i$, to get its conditional posterior, we first remove all the words in document $i$ out of the statistics, then add them in one by one again following their topic assignments $\{z_{il}\}$. This would cause a likelihood change, *i.e.*, $\prod_{l=1}^{N_i} p\left(z_{il} | y_i, \text{others}\right)$ in (12). Another term what would change in (12) is the term $\frac{\prod_{k=1}^{K} \Gamma(C_k + \omega)}{\Gamma(C_\cdot + \omega K)}$.

Finally, combining the terms in $\tilde{\phi}_i(\lambda_i|\Theta)$ by the max-margin constraints, we get the joint conditional posterior of $(y_i, s_i)$ as:

$$q(y_i = k, s_i = \tau(k)|\text{others}) \propto (C_k + \omega)f(y_i|\Theta) \,,$$

where $\tau(k) = \arg\max_{k':k'\neq k} \zeta_{ik'}$, $f(y_i|\Theta)$ is the pseudo likelihood defined as

$$f(y_i|\Theta) := \prod_{l=1}^{N_i} p(z_{il}|y_i, Z_{<(i,l)}, \text{others}) \times$$
$$\exp\left\{ -\frac{1}{2\lambda_i}\left( \lambda_i + c\left( \ell - (\boldsymbol{\eta}_{y_i} - \boldsymbol{\eta}_{s_i})^T \mathbf{x}_i \right) \right)^2 \right\} \,,$$

where the subscript $< (i, l)$ means the statistics up to word $l$ in document $i$ because here the words are added in sequentially.

# D   Additional results

## D.1   Influence of NMI scores by hyperparameters in Bayesian nonparametric cluster models

We demonstrate how the hyperparameters affect the NMI scores in two Bayesian nonparametric clustering models in Figure 6 and Figure 7.

Figure 6: Influence of NMI score by hyparameter $r$ (see Figure. 1 (left) in the main text for the definition) in DPMMGM [10] on the *Wheat Kernel* dataset used in [35] ( the whole dataset). Other hyperparameters (refer to Figure. 1 (left) in our paper) are set as $\mathbf{S} = I_7, \mathbf{m} = \mathbf{0}, \nu = 37$, which are empirically performed well. We can see that the NMI performance varies significantly with respect to $r$. It was not optimal even if we sampled it by placing another hierarchical prior on it.

## D.2   CPMMC vs. DPMMGM

In this section we compare the max-margin clustering model [7] with our DPMMGM in Table 3.

## D.3   Cluster embedding

For better view of the document clustering embedding results with the MMCTM, Figure 8 re-plots the figures in a larger size, from which we can see clearly that MMCTM clusters documents by inducing larger margins between clusters.

Figure 7: Influence of NMI score by hyparameter $L$ (see Section 2.1.4 in [35]) in DCP [35] on the same dataset using the code from the authors. We varied the $L$ hyperparameter between 0.1 to 3, and sampled all the other other hyperparameters. Again we can see that the NMI performance varies significantly with respect to $L$. It was not optimal even if we sampled it (the red curve).

| Datasets | CPMMC | DPMMGM |
|----------|-------|--------|
| Digits | 0.5350 | **0.5386** |
| Glass | 0.0693 | **0.1173** |
| Half_cicle | 0.4291 | **0.4726** |
| Iris | 0.0186 | **1** |
| Letter | 0.2008 | **0.6987** |
| MNIST | **0.9229** | 0.8506 |
| Satimage | 0.8558 | **1** |
| Segment'n | N/A | **1** |
| Vehical | 0.0607 | **0.0626** |
| Vowel | 0.0044 | **0.0129** |
| Wine | 0.8428 | **0.9128** |

Table 3: Comparison of CPMMC [7] and DPMMGM in term of NMI scores. For each dataset, we randomly chose 2 classes out of the datasets. The first dataset is the one used in the demonstration of their code by the authors, the rest are the datasets used in our paper. For each dataset, related parameters of CPMMC were manually tuned so that it obtained the best NMI scores. "N/A" in the table means infinite loop for the dataset thus no result produced. Obviously our DPMMGM outperforms CPMMC.

## D.4 T-test on the clustering results of SVM and MMCTM

To further illustrate the differences between our proposed MMCTM model, we do t-test with the state of the art non-probilistic model – SVM. The results are shown in Table 4, from which we can see that MMCTM* with optimal parameter settings is mostly significantly better than SVM, while MMCTM with fixed hyperparameters is at least as good as SVM, and in some cases it is significantly better.

## D.5 Convergence speed

This section shows the convergence speed of the MMCTM in term of clustering accuracy on the two datasets. Figure 9 plots the curves on the *20NEWS* dataset, and Figure 10 plots the curves on the *Reuters-R8* dataset. The results are means and standard derivations over 5 repeated runs with random

(a) MMCTM on *20NEWS*

(b) CTM on *20NEWS*

(c) MMCTM on *Reuters-R8*

(d) CTM on *Reuters-R8*

Figure 8: Clusters embedded in 2-dimensional space with tSNE. Different colours correspond to different clusters. Best view in colour.

initialisations. We can see that in general the accuracies converge after 30 iterations, demonstrating the efficiency of our sampler.

## D.6 Topic illustrations

In this section we show 5 clusters and their associated top 5 topics in each cluster on the *Reuters-R8* dataset in Table 5. We can see from the table that in general each cluster corresponds to specific theme, with several junk topics consists of stop words. Note we did not remove the stop words, and believe it performs even better after removing them.

| $L$ | SVM | MMCTM | MMCTM* |
|---|---|---|---|
| *20NEWS* | | | |
| 5 | $37.13_{\pm 2.9}$ | $\mathbf{56.70}_{\pm 1.9}$ $(h=1, p=0.0006)$ | $\mathbf{57.86}_{\pm 0.9}$ $(h=1, p=0.0001)$ |
| 10 | $46.99_{\pm 2.4}$ | $\mathbf{54.92}_{\pm 1.6}$ $(h=1, p=0.0102)$ | $\mathbf{56.56}_{\pm 1.3}$ $(h=1, p=0.0018)$ |
| 15 | $52.80_{\pm 1.2}$ | $55.06_{\pm 2.7}$ $(h=0, p=0.1962)$ | $\mathbf{57.80}_{\pm 2.2}$ $(h=1, p=0.0152)$ |
| 20 | $56.10_{\pm 1.5}$ | $56.62_{\pm 2.2}$ $(h=0, p=0.7057)$ | $\mathbf{59.70}_{\pm 1.4}$ $(h=1, p=0.0347)$ |
| 25 | $59.15_{\pm 1.4}$ | $55.70_{\pm 2.4}$ $(h=0, p=0.0668)$ | $\mathbf{61.92}_{\pm 3.0}$ $(h=1, p=0.0456)$ |
| *Reuters-R8* | | | |
| 5 | $78.12_{\pm 1.1}$ | $79.18_{\pm 4.1}$ $(h=0, p=0.6597)$ | $80.86_{\pm 2.9}$ $(h=0, p=0.1943)$ |
| 10 | $80.69_{\pm 1.2}$ | $80.04_{\pm 5.3}$ $(h=0, p=0.8265)$ | $\mathbf{83.48}_{\pm 1.0}$ $(h=1, p=0.0341)$ |
| 15 | $83.25_{\pm 1.7}$ | $85.48_{\pm 2.1}$ $(h=0, p=0.0683)$ | $86.86_{\pm 2.5}$ $(h=0, p=0.0823)$ |
| 20 | $85.66_{\pm 1.0}$ | $82.92_{\pm 1.7}$ $(h=0, p=0.0742)$ | $83.82_{\pm 1.6}$ $(h=0, p=0.1711)$ |
| 25 | $84.95_{\pm 0.1}$ | $86.56_{\pm 2.5}$ $(h=0, p=0.2213)$ | $\mathbf{88.12}_{\pm 0.5}$ $(h=1, p=0.0005)$ |

Table 4: T-test results for the best non-probalistic algorithm SVM and the proposed MMCTM. $h=1$ means significantly different from SVM, while $h=0$ means not significantly different, $p$ is the corresponding $p$-value in the t-test. **Bold** means significantly better.

(a) $L=5$

(b) $L=10$

(c) $L=15$

(d) $L=20$

(e) $L=25$

Figure 9: Accuracy vs. the number of iterations for MMCTM on the *20NEWS* dataset. $L$ is the number of landmarks.

(a) $L = 5$
(b) $L = 10$
(c) $L = 15$
(d) $L = 20$
(e) $L = 25$

Figure 10: Accuracy vs. the number of iterations for MMCTM on the *Reuters-R8* dataset. $L$ is the number of landmarks.

| | | | | | Cluster 1: tax | | | | | |
|---|---|---|---|---|---|---|---|---|---|---|
| T_1 | mln | cts | Net | loss | shr | Reuter | dlrs | qtr | year | revs |
| T_2 | dlrs | share | for | cts | and | per | quater | company | earnings | results |
| T_3 | year | and | dlrs | for | company | will | quater | mln | pct | earnings |
| T_4 | mln | billion | profit | Net | dlrs | loss | tax | shr | pretax | extraordinary |
| T_5 | cts | Net | shr | loss | mln | revs | profit | qtr | Reuter | inc |
| | | | | | Cluster 2: stock | | | | | |
| T_1 | for | and | offer | dlrs | share | company | that | will | not | inc |
| T_2 | and | shares | pct | company | stock | group | dlrs | Reuter | common | stake |
| T_3 | and | corp | sale | unit | mln | inc | Reuter | will | company | dlrs |
| T_4 | that | analysts | not | market | analyst | stock | American | spokesman | comment | last |
| T_5 | general | and | gencorp | inc | for | partners | dlrs | taft | mln | acquire |
| | | | | | Cluster 3: trade | | | | | |
| T_1 | that | and | for | not | will | told | this | had | meeting | world |
| T_2 | trade | and | reagan | house | bill | countries | gatt | congress | talks | states |
| T_3 | Japan | Japanese | trade | officials | pact | united | Washington | tariffs | semiconductor | last |
| T_4 | trade | billion | exports | dlrs | and | surplus | imports | deficit | Taiwan | year |
| T_5 | and | foreign | for | companies | government | industry | year | sources | this | years |
| | | | | | Cluster 4: agriculture | | | | | |
| T_1 | grain | Soviet | agriculture | farm | tonnes | usda | certificates | mln | corp | and |
| T_2 | that | and | for | not | will | told | this | had | meeting | world |
| T_3 | year | and | dlrs | for | company | will | quarter | mln | pct | earnings |
| T_4 | and | foreign | for | companies | government | industry | year | sources | this | years |
| T_5 | trade | and | reagan | house | bill | countries | gatt | congress | talks | states |
| | | | | | Cluster 5: bank | | | | | |
| T_1 | dollar | exchange | currency | and | baker | west | Paris | rates | accord | German |
| T_2 | that | and | for | not | will | told | this | had | meeting | world |
| T_3 | bank | stg | mln | market | money | bills | England | pct | today | Reuter |
| T_4 | and | markets | budget | government | dollar | that | economy | policy | exchange | prices |
| T_5 | and | foreign | for | companies | government | industry | year | sources | this | years |

Table 5: Top 5 topics in each cluster.

## Auxiliary References

[39] R. M. Neal. Slice sampling. *Ann. Statist.*, 31(3):705–767, 2003.

[40] L. Devroye. Random variate generation for the generalized inverse Gaussian distribution. *Stat. Comput.*, 2012.

[41] R. M. Neal. Markov chain sampling methods for Dirichlet process mixture models. *J. Comput. Graph. Stat.*, 2000.