[Reviews · NeurIPS 2014]

Submitted by Assigned_Reviewer_9

The paper introduces max-margin Bayesian clustering (BMC) that extends Bayesian clustering techniques to include the max-margin criterion. This includes, for example, the Dirichlet process max-margin Gaussian mixture that relaxes the underlying Gaussian assumption of Dirichlet process Gaussian mixtures by incorporating max-margin posterior constraints, and is able to infer the number of clusters from data. The resulting techniques (DPMMGM and a further one classed MMCTM) are compared to a variety of other techniques in several numerical experiments.

The paper combines two clustering approaches: Deterministic and Bayesian clustering. Deterministic approaches can easily handle constraints, but lack the ability to infer the number of clusters. Bayesian techniques can infer the number of clusters, but not handle constraints such as the max-margin.

Quality:

The resulting BMC algorithm is claimed to be the first extension of regularized Bayes inference to the unsupervised clustering task. Furthermore, the DPMMGM algorithm uses the max-margin constraints to relax the Gaussian assumptions of the standard DPGMM, which is a strong new capability.

Clarity:

(pg 4, 184) “The max-margin constraints take effects in the model via \tilde\phi_i’s in (8).” Perhaps this claim was better illustrated in earlier work, but why this statement is true is not made sufficiently clear in this paper. The authors need to provide better support for these claims.

(pg 4, 208) “Note that DPMMGM does not need the complicated class balance constraints [6] because the Gausses in the pseudo likelihood would balance the clusters to some extent.” Not sure that “Gausses” is an appropriate wording. This statement ending of “to some extent” is not sufficiently precise for this type of publication.

(pg 6, 298) controll is misspelled

(pg6, 284) Fig 2 shows that the two algorithms generate different results, but the discussion provided here does not give sufficient indication of why the result of the DPMMGM is better. DPGMM is claimed to be more fragmented, but why is that significant here, and, if anything, the DPGMM appears to be a better fit to the data, so what has been lost in switching to the DPMMGM?

(pg6, 295) “thus driving the data points to collapse as well.” Pls clarify this statement. Please also expand this discussion to connect the intuition to the figure 3 (a)-(j). There are a lot of figures, a lot of trends, and each figure is difficult to interpret with the lines and clusters. The conclusion “the results indeed follow our intuition,” is far less obvious than the authors claim.

Originality:

While it would appear that the ideas are original, much of section 3 on robust Bayesian max-margin clustering follows [17] very closely. In particular the key steps between equations (6)-(7)-(8) reuse the techniques introduced in [17]. This suggests the possibility that the core ideas are in the prior work, and the authors are just re-applying them here to a new model formulation, which calls into question the novelty and importance of this work. Without further clarification, this influences the impact score.

Significance:

The extensive numerical results show that DPMMGM outperforms DPGMM, and similarly with MMCTM vs. SVM and S3VM, which appear to be significant results.

However, while the paper contains a lot of material on model development, there appear to be very few theoretical statements on the performance or convergence of the algorithms.
Summary: Interesting paper topic. Resulting algorithms do well in the numerical comparisons, that seem extensive. But further clarification compared to the technical approach in [17] is required.

Submitted by Assigned_Reviewer_23

The authors propose a clustering framework in which max-margin constraints are invoked such that clusters are defined as a tradeoff between modeling the data density and separating observations in terms of cluster specific latent projections based on regularized Bayesian inference. The method is derived for standard DPGMM as well as topic modeling based on the cluster based topic model (CTM) forming the proposed DPMMGMM and MMCTM and the utility of the approach demonstrated on a synthetic dataset as well as several real datasets demonstrating that the ground truth information is better recovered when compared to standard non-parametric mixture modeling (DPGMM and cluster based topic modeling, CTM) as quantified respectively in terms of normalized mutual information and accuracy.

The framework is interesting and invoking separation of clusters is both reasonable and seems to provide significant utility which may warrant publication. The proposed framework extends the regularized Bayesian inference framework and the non-trivial implementation details are provided in the accompanying appendix.

My main criticism has to do with the tuning of the regularization strength c and margin l. Clearly as also discussed in the paper these parameters have great influence on the results. However, their suggested tuning is a bit ad hoc and it would improve the paper to inspect their cross-validated tuning versus the proposed heuristic forming DPMMGMM*. Furthermore, it is unclear what denotes the MMCTM* framework it is just stated in the paper that this constitutes the optimal parameter setting - but how was this found (do you mean using the same number of topics as for CTM - Please clarify)? For the topic modeling results in particular it seems that the choice of c=9 and l=0.1 is a rather informed setting and it is unclear how this setting was selected. This should also be clarified or at least the influence of these choices be better accounted for as the tuning of these parameters for the approach seems to be somewhat of a key issue for performance.

Minor comments:
Please clarify what the lines denote in Figure 3. I assume they are representing \eta_k.

Gausses -> Gaussians
in the generate process -> in the generative process
needed to sampled -> needed to be sampled
to jointly sampled (y_i,s_i) -> to jointly sample (y_i,s_i)
many of the the sampling formula are -> many of the sampling formulas are

Summary: An interesting framework for separating clusters in generative models that appears to have utility in extracting ground truth structure. However, adequate tuning of tradeoff and marging parameters seems to be essential.

Submitted by Assigned_Reviewer_28

This paper presents a fully Bayesian framework which enables max-margin constraints to be incorporated for clustering models. The method leverages the existing Regularized Bayesian inference technique from Zhu et al. but extends it to unsupervised settings. The authors demonstrate the efficacy of the new approach using two different clustering models: a DP- based GMM and a cluster-topic model, both with integrated max-margin constraints.

The paper is well written with clear description and details of the new max-margin clustering approach and the model variants. It would be useful if you can expand on the posterior inference part a bit more (more details on sampling complexity) since practical performance highly depends on the efficiency of the sampler. The authors also compare the two models to baselines systems and analyze sensitivity to the hyper parameters, which is nice.

Instead of using the the heuristic for model selection, would it be possible to make it non-parametric & estimate this during inference (perhaps your heuristic could be used as an informative prior)?

Clarification: Is “p” the dimensionality of the latent space? So the time complexity for DPGMM has an extra cubic factor in p.

It is encouraging to see that adding max-margin constraints improves clustering quality when using unsupervised Bayesian models. The datasets in Table 1 are still small and I wonder if the new method can scale to practical settings when the data & dimensionality is really large. Perhaps some of the recent work on fast sampling using randomization and other approximation techniques could be used to speed up inference further. It will be nice to address this somewhere in the paper (either as future work or in related work).

Are the accuracy differences in Table 2 statistically significant (e.g., S3VM vs. MMCTM for L=5,10)? Please provide stat sig. numbers.

Summary: The paper is well written and extends the RegBayes framework to unsupervised settings. The authors demonstrate the utility of the new approach with two different clustering models showing that adding max-margin constraints improve clustering quality.
Author Feedback
Author rebuttal: We thank the reviewers for their valuable comments. We will improve the paper accordingly. Below we address the detailed comments.

To Reviewer_23:

Q: main criticism has to do with the tuning of the regularization strength c and margin l.
A: we agree that tuning the hyperparameters c and l with cross validation (CV) could be beneficial, but we didn't do it because we didn't separate the data for training and testing respectively in the clustering tasks. We actually had tried CV on some other datasets offline and found that it was comparable to our heuristic approach used. Furthermore, our method saves computational time. So we believe it is practical. We will discuss this in more details in the revision.

Q: unclear what denotes the MMCTM*? Model selection for MMCTM?
A: For MMCTM, we found that the performance is not as sensitive to c and l as DPMMGM does (it might be because the margin constraints are put on the latent space). So we just fixed c=9 and l=0.1 in all the experiments. Sorry for the confusion, MMCTM* is the method whose c and l correspond to the best performance found by grid search on c and l, thus can be roughly considered as the optimal model. We showed that with the chosen c and l, MMCTM performs almost as well as MMCTM*, thus is robust. We believe choosing them with CV is also a good way.

Yes, the lines in Figure 3 denote \eta_k.

To Reviewer_28:

Q: would it be possible to make it non-parametric?
A: About using nonparametric method for model selection, our recent finding is that it doesn't work. The reason is that by inspecting the posterior formula, we find that the smaller the c is, the larger the model likelihood is, so c would always tend to 0 when sampling. The l parameter, however, could be sampled when fixing c, but it will increase the computational cost significantly. Thus we chose not to sample at the current stage. It is interesting to find efficient ways for the sampling in the future.

Q: Is “p” the dimensionality of the latent space?
A: Actually p is the dimension of the data space. We didn't introduce latent space in DPMMGM. This is why the model could not handle high dimensional data. However, this can be solved by putting the max-margin constraint on a latent space (e.g., by projection), like what MMCTM does. Investigating other latent space learning methods is an interesting future direction.

Q: Are the accuracy differences in Table 2 statistically significant?
A: The accuracy differences in Tab.2 are significant. We will provide the sig. numbers in the revision.

To Reviewer_9:

Sorry for the confusions. We will improve the clarity in revision.

Q: Clarity:
A: (pg 4, 184): the max-margin takes effects via \tilde\phi_i, this can be seen from the derivation of eq.8, which shows that the constraints have been absorbed into \phi_i's. We will make it clearer in revision.

(pg 6, 284): we want to emphasize here that although DPGMM appears to fit the data better, it doesn't necessary generate better clustering results. In our case, each cluster is a half cycle and not Gaussian, our method can handle this with max-margin constraints added while DPGMM fails because of the Gaussian assumption.

(p6, 295): larger l driving data points to collapse means that large l doesn't allow any constraint to be violated because it will generate large loss, thus all the data points tend to be in a single cluster where no margin constraints are needed. Figure 3 essentially shows that the number of clusters decreases with increasing c and l. We will make this clearer in revision.

We agree the paper is based on RegBayes. However, as emphasized in Section 3, there are key technical challenges and novel solutions to call for novel solutions by generalizing RegBayes to a clustering framework. For example, we generalize it to unsupervised setting, and introduce additional auxiliary variables s_i's to make the model applicable to mutil-class case, which is essential for clustering models. These make it significantly different from existing RegBayes methods.

Finally, despite the lack of theoretical analysis, the convergence is empirically tested in appendix, which shows fast convergence rates for our models. We agree some theoretical analysis on the model is interesting future work.